# MACHINE UNLEARNING FOR ALLEVIATING NEGATIVE TRANSFER IN PARTIAL-SET SOURCE-FREE UNSUPERVISED DOMAIN ADAPTATION

## ABSTRACT

Source-free Unsupervised Domain Adaptation (SFUDA) aims to adjust a source model trained on a labeled source domain to a related but unlabeled target domain without accessing the source data. Many SFUDA methods are studied in closed-set scenarios where the target domain and source domain categories are perfectly aligned. However, a more practical scenario is a partial-set scenario where the source label space subsumes the target one. In this paper, we prove that reducing the differences between the source and target domains in the partial-set scenario helps to achieve domain adaptation. And we propose a simple yet effective SFUDA framework called the Machine Unlearning Framework to alleviate the negative transfer problem in the partial-set scenario, thereby allowing the model to focus on the target domain category. Specifically, we first generate noise samples for each category that only exists in the source domain and generate pseudo-labeled samples from the target domain. Then, in the forgetting stage, we use these samples to train the model, making it behave like the model has never seen the class that only exists in the source domain before. Finally, in the adaptation stage, we use only the pseudo-labeled samples to conduct self-supervised training on the model, making it more adaptable to the target domain. Our method is easy to implement and pluggable, suitable for various pre-trained models. Experimental results show that our method can well alleviate the negative transfer problem and improve model performance under various target domain category settings.

## 1 INTRODUCTION

Although computer vision tasks have achieved significant success in various fields (Voulodimos et al., 2018), current deep models often experience a decline in performance when tested in new environments that have a considerable domain gap from the training environment (Patel et al., 2015). Unsupervised Domain Adaptation (UDA) successfully addresses model performance degradation by automatically learning transformations of the feature space, allowing knowledge from the source domain to be transferred to the target domain (Wilson & Cook, 2020). However, traditional UDA methods require datasets and labels from the source domain (Ganin et al., 2016). Firstly, the dataset from the source domain is likely to be quite large (1∼100 GB) (Lin et al., 2014; Yu et al., 2020), which presents significant challenges in terms of storage and transmission. Secondly, many datasets are not publicly available due to privacy and security considerations (Sun et al., 2017; Tian et al., 2023; Nagrani et al., 2018). In many cases, we can only access models trained in the source domain, which has led to widespread interest in Source-Free Unsupervised Domain Adaptation (SFUDA).

SFUDA aims to transfer a pre-trained model from the source domain to the target domain without source data (Fang et al., 2024). Due to the lack of images and labels from the source domain, SFUDA cannot directly employ the methods used in UDA, such as adversarial learning (Goodfellow et al., 2014; Ganin et al., 2016). In addition, most SFUDA methods focus on closed-set scenarios, where the source and target domain samples come from the same category set (Fang et al., 2024; Liang et al., 2024). But a more realistic scenario is the partial-set shown in Figure 1, where some source domain categories do not exist in the target domain. The mismatch in class labels can lead to the problem of negative transfer, making domain adaptation tasks more difficult. Unlike in the UDA

setting, where specific source domain samples can be selected for re-training the model based on the target domain class configurations (Guo et al., 2022). In the SFUDA setting, the negative transfer problem is more challenging. However, very few SFUDA methods works in partial-set scenarios. Although approaches like SHOT(Liang et al., 2020) and HCT(Huang et al., 2021) demonstrate good generalization capabilities under partial-set, they do not specifically address the negative transfer problem caused by the class mismatch between the source domain and the target domain.

Most SFUDA methods are inspired by semi-supervised learning, and a typical SFUDA method is to train models using pseudo-labels from the target domain (Chien et al., 2023; Liang et al., 2024). However, due to domain shift (Moreno-Torres et al., 2012), these pseudo-labels are often noisy, which can lead to confirmation bias (Yang et al., 2022), significantly affecting the performance of the model. Yang et al. (2022) introduces subdomain augmentation in twin network teaching, effectively solving the problem; Zhang et al. (2022) avoids the influence of incorrect labels through a new paradigm of separation by specialized learning. However, they all significantly increase the complexity of the method. Another popular approach is to fill in the missing data of the source domain, which helps turn challenging SFUDA problems into well-studied UDA problems (Fang et al., 2024). For example, Kurmi et al. (2021); Li et al. (2020) train a GAN-based generator to simulate the source data; Ding et al. (2023); Ye et al. (2021); Du et al. (2024) adopt proxy source data construction, where suitable samples are directly selected from the target domain to replace the source data. However, these methods may not effectively represent the original source domain.

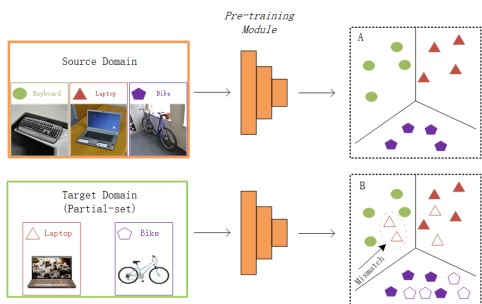

Figure 1: Illustration of partial-set scenario. In the source domain, we have three categories: Keyboard, Laptop and Bike. We obtain a pretrained model and its classification results (denoted as A) on the source domain. In the target domain, which has a partial dataset scenario, only two categories — Laptop and Bike — are present. We obtain the classification results (denoted as B) on the source domain. It can be observed that the negative transfer problem caused by partial-set scenarios leads to the occurrence of mismatches.

To this end, we propose a Machine Unlearning framework aimed at solving the negative transfer problem caused by the mismatch of category information between the source and target domains, thereby achieving better performance of the model in the target domain. The framework consists of two main stages: in the forgetting stage, we use a pre-trained model to make predictions on the target domain data, selecting reliable samples as a pseudo-label dataset based on the model's confidence in its predictions. Simultaneously, we generate noise samples for each category that only exists in the source domain to create a noise sample dataset. We use them to train the model, allowing it to forget the information of each redundant category in the source domain while minimizing the impact on other categories. In the adaptation stage, we use the reliable pseudo-label dataset for self-supervised learning of the model, allowing the model to adapt more effectively to the target domain. The parameters in our framework are easy to adjust, and training the model does not require high performance hardwares, which makes our approach easy to implement. The experiment prove that our methods effectively solve the negative transfer problem and improve the performance of the model in the target domain under multiple target domain category settings. Overall, the main contributions of this work are described as follows:

- We study the partial-set scenario in SFUDA and prove that reducing the differences between the source and target domains in the partial-set scenario helps to achieve domain adaptation.

- We innovatively introduce machine unlearning into SFUDA, and design an efficient and easy-to-implement framework. Furthermore, our method is pluggable.

- The experimental results show that our work can effectively alleviate the negative transfer problem in partial-set scenario and improve the accuracy of the model on target data under various target domain category settings.

## 2 RELATED WORK

**Unsupervised Domain Adaptation.** Due to the differences between domains, classifiers trained on the source domain may experience performance degradation when tested on the target domain. UDA specifically addresses the situation where target data are not labeled in domain adaptation (Ganin et al., 2016; Yang et al., 2024), eliminating the reliance on potentially expensive target data labels in domain adaptation. The main UDA methods focus on learning domain-invariant features, with the aim of aligning the characteristics of the target domain data with those learned during model training (Zhao et al., 2019). In addition, there are many studies that address the partial-set problem in UDA, such as Guo et al. (2022) uses maximum cosine similarity (MoC) to select useful data in the source domain for retraining, in order to reduce domain differences; Wang & Breckon (2021) uses Locally Preserved Projection (SLPP) to better align two domains in the subspace, and detects the source domain category of non target domain categories, deletes them, and retrains the model. However, UDA relies too heavily on source domain data, whereas SFDUA is more practical.

**Source-free Unsupervised Domain Adaptation.** SFUDA is a domain adaptation method that uses only the source domain model without using the data of the source domain. Recently, an increasing number of SFUDA methods have been applied in fields such as image classification, object recognition, and face anti-spoofing (Liu et al., 2021; Chen et al., 2022; Liu et al., 2022). The main SFUDA research includes methods for generating source domain data and fine-tuning models (Fang et al., 2024), and even studies that use API services of source domain models for knowledge distillation (Yang et al., 2022; Liang et al., 2022). While the aforementioned methods demonstrate strong performance, most studies are constrained to closed-set scenarios. In practice, it is rare for the source domain and the target domain to completely share the label space, and the labels in the target domain usually only comes from a subset class of the source domain. In the partial-set scenario, directly using traditional transfer methods cannot address the issue of negative transfer, which leads to a decline in the model's performance. Thus, we are committed to solving the negative transfer problem caused by mismatched label spaces between source and target domains, and achieving more effective knowledge transfer.

**Machine Unlearning.** The goal of Machine Unlearning is to replicate a model that consumes less time and performs consistently with models trained without specific data (Nguyen et al., 2022; Wang et al., 2023; Li et al., 2024). This is a special requirement arising from privacy, availability, and the right to be forgotten (Dang, 2021). In fact, removing the influence of abnormal training samples from the model can also lead to higher model performance and robustness (Chien et al., 2023; Wang et al., 2023). The methods of Machine Unlearning can be roughly divided into data reassembly methods and weight model manipulation methods. For example, SISA(Bourtoule et al., 2021) partitions and sorts data, retraining only the model for the partition containing the forgotten data; Task Arithmetic(Ilharco et al., 2023) modifies pre-trained model behavior via task vectors, enabling significant performance reduction in the targeted task through task vector subtraction while minimally affecting other tasks. We use the mechanism of Machine Unlearning to eliminate categories that only exist in the source domain, addressing the negative transfer problem in artial-set scenarios and thereby enhancing the model's performance in the target domain.

## 3 METHOD

In this section, we first describe the partial-set scenario task under source-free unsupervised domain adaptation. We then prove that reducing the disparity between the source and target domains in the partial-set scenario facilitates domain adaptation. Next, we elaborate on our proposed method, the Machine Unlearning Framework, which consists of three steps: acquiring the sample set needed for model updating, the forgetting stage and the adaptation stage. Specifically, the framework utilizes a filtering mechanism to filter the pseudo-labels in the target domain. It then generates noise samples for each category unique to the source domain to facilitate model forgetting. Finally, it employs pseudolabels from the target domain for simple self-supervised learning. In particular, our framework is capable of iterative updates, enabling the model to achieve stronger performance.

To provide a visual overview of our methodology, we present Figure 2 to illustrate the entire framework.

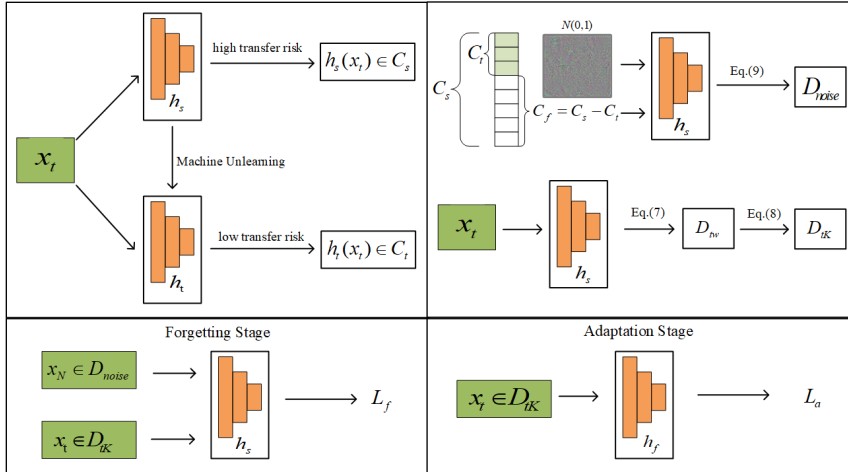

Figure 2: Illustration of Machine Unlearning Framework. In partial-set scenario, directly transferring the pre-trained model $h_s$ to predict the target domain data $x_t$ carries a high risk. To reduce the transfer risk, we obtain model $h_t$ through our Machine Unlearning framework. Specifically, we first generate a noise sample dataset $D_{noise}$ for each category in the unique category set $C_f$, and use a pre-trained model $h_s$ to obtain a pseudo labeled dataset $D_{tK}$. Then in the forgetting stage, we minimize $L_f$ on $h_s$ to obtain the forgetting model $h_f$. Finally, we minimize $L_a$ on $h_f$ to obtain the target model $h_t$ in the adaptation stage.

## 3.1 PROBLEM SETUP

In SFUDA settings, we have a labeled source domain dataset $D_s = \{(x_s^i, y_s^i)\}_{i=1}^{n_s}$ and an unlabeled target dataset $D_t = \{(x_s^i)\}_{i=1}^{n_t}$, where $n_s$ and $n_t$ represent the number of samples in the source and target domains, respectively. The source domain model trained in $D_s$ is defined as $h_s$. SFUDA aims to predict the labels of the target domain using the target model trained from the source domain model. Similarly, we define the target domain model as $h_t$. We define the sets of source domain and target domain categories as $C_s$ and $C_t$, respectively. The current deep models are trained on large-scale datasets such as ImageNet-1K (Deng et al., 2009). We consider partial set scenarios that are more realistic than closed set scenarios, and we have $C_t \subseteq C_s$.

In classification tasks, the model not only outputs the predicted class for the image but also provides the score associated with the predicted class. We define the scoring function of the source domain model $h_s$ in the hypothesis space $F$ as $f_s : x \to \mathbb{R}^{C_s}$ where the output on each dimension represents the predicted score for that category. We know that the classification model will output the category with the highest score. We define the prediction category of $h_s$ for $x$ as $y = h_s(x)$, with a little abuse of notation, we consider $f_s(x, y)$ is the score corresponding to the prediction category of $h_s$ for $x$. Similarly, we define the scoring function of the target domain model $h_t$ as $f_t : x \to \mathbb{R}^{C_t}$, and we consider $f_t(x, y)$ is the score corresponding to the prediction category of $h_t$ for $x$.

If we directly apply the traditional SFUDA method to obtain the model $h_{tr}$, the corresponding scoring function is $f_{tr} : x \to \mathbb{R}^{C_s}$. This model may output only the classes presented in the source domain, which leads to the negative transfer issue. Therefore, we focus on reducing the discrepancy between the source and target domains in partial-set scenarios through Machine Unlearning, thereby alleviating the negative transfer issue associated with domain adaptation and ultimately improving the model's performance in the target domain. Next, in order to demonstrate that reducing the discrepancy between the source and target domains in partial-set scenarios is helpful for domain adaptation, we provide a theoretical analysis of model transfer errors.

## 3.2 THEORETICAL ANALYSIS OF MODEL TRANSFER ERRORS

The error rate of model $h$ on dataset $D$ is given by :

$$\text{err}_D(h) \triangleq \mathbb{E}_{(x,y)\sim D} \mathbf{1}[h(x) \neq y], \tag{1}$$

where $\mathbf{1}$ represents the indicator function.

In practice, the boundary between the data samples and classification plays an important role in achieving strong generalization performance. Therefore, Koltchinskii & Panchenko (2002) proposes a marginal theory for classification, in which the 0-1 loss is replaced by the Margin Loss.

**Definition 1. Margin Loss.** $f$ is the scoring function of $h$. We define the margin of the sample $(x, y)$ for $f$ as $\varrho_f(x, y) = \frac{1}{2}(f(x, y) - \max_{y' \neq y} f(x, y'))$. We set $G_\varrho(x) = \begin{cases} 0, & \varrho \leq x \\ 1 - \frac{x}{\varrho}, & 0 \leq x \leq \varrho \\ 1, & x \leq 0 \end{cases}$.

The Margin Loss is then defined as:

$$err_D^{(\varrho)}(f) \triangleq \mathbb{E}_{x \sim D} G_\varrho(\varrho_f(x, y)). \tag{2}$$

An important property is that $err_D^{(\varrho)}(f) \geq err_D(h)$ for any $\varrho > 0$. This property is important for multi-class classification task since it can better reflect the quality of the model. Based on the margin loss, we introduce the margin disparity to compare the prediction differences between any two models on the same dataset.

**Definition 2. Margin disparity.** Given two models $h, h' \in H$, their scoring functions are $f$ and $f'$, respectively. The Margin Disparity is defined as:

$$\mathrm{disp}_D^{(\varrho)}(f', f) \triangleq \mathbb{E}_{x \sim D} G_\varrho(\varrho_{f'}(x, h(x))), \tag{3}$$

where the value of $\mathrm{disp}_D^{(\varrho)}(f', f)$ is a real number between 0 and 1.

In domain adaptation, the model transfer errors often depend on the distribution differences between the source and target domains. Therefore, we introduce the following distribution difference metric.

**Definition 3. Margin Disparity Discrepancy(MDD).** Based on the margin disparity and referring to Zhang et al. (2019), we further provide the margin disparity discrepancy to measure the difference between the source domain and the target domain.

$$d_{f,F}^{(\varrho)}(D_s, D_t) \triangleq \sup_{f' \in \mathcal{F}} (\mathrm{disp}_{D_t}^{(\varrho)}(f', f) - \mathrm{disp}_{D_s}^{(\varrho)}(f', f)). \tag{4}$$

When $D_s$ and $D_t$ are equal, $d_{f,F}^{(\varrho)}(D_s, D_t) = 0$. Although MDD does not satisfy symmetry, it has nonnegativity and subadditivity, so it can measure the distribution differences between the source and target domains. Now we define $\tilde{D}_s = \{(x_s^i, y_s^i) | (x_s^i, y_s^i) \in D_s, y_s^i \in C_t\}_{i=1}^{\tilde{n}_s}$, where $\tilde{n}_s$ is the number of samples in $C_t$ from $D_s$. The model trained on $\tilde{D}_s$ is represented as $\tilde{h}_s$, and the corresponding scoring function is $\tilde{f}_s$. We can obtain a new MDD about $\tilde{D}_s$ and $D_t$, denoted as $d_{f,F}^{(\varrho)}(\tilde{D}_s, D_t)$. However, considering that $\mathrm{disp}_{D_s}^{(\varrho)}(f', f)$ and $\mathrm{disp}_{\tilde{D}_s}^{(\varrho)}(f', f)$ are difficult to obtain, it is difficult to directly compare $d_{f,F}^{(\varrho)}(D_s, D_t)$ and $d_{f,F}^{(\varrho)}(\tilde{D}_s, D_t)$.

**Theorem 1.** For the target domain model $h_t$,

$$err_{D_t}(h_t) \leq err_{D_s}^{(\varrho)}(f_t) + \sup_{f' \in \mathcal{F}} (disp_{D_t}^{(\varrho)}(f', f_s)) - \inf_{f' \in \mathcal{F}} (disp_{D_s}^{(\varrho)}(f'', f_s)) + \lambda, \tag{5}$$

where $\lambda = \min_{f^* \in \mathcal{F}} \{err_{D_s}^{(\varrho)}(f^*) + err_{D_t}^{(\varrho)}(f^*)\}$ is independent of $h_t$. Both $f'$ and $f''$ are trained by $f_t$ using $D_t$ and $D_s$.

Proof. Please refer to Appendix A.1.

In Theorem 1, we obtain an upper bound of $err_{D_t}(h_t)$. Specifically, we decompose MDD into supremum and infimum components to resolve the difficulty in directly comparing the magnitudes of $d_{f,F}^{(\varrho)}(D_s, D_t)$ and $d_{f,F}^{(\varrho)}(\tilde{D}_s, D_t)$.

**Theorem 2.** In the same hypothesis space $F$, $h_s$ is trained on $D_s$, and $\tilde{h}_s$ is trained on $\tilde{D}_s$,

$$\sup_{\tilde{f}_s' \in \mathcal{F}} (\mathrm{disp}_{D_t}^{(\varrho)}(\tilde{f}_s', \tilde{f}_s)) - \inf_{\tilde{f}_s'' \in \mathcal{F}} (\mathrm{disp}_{\tilde{D}_s}^{(\varrho)}(\tilde{f}_s'', \tilde{f}_s)) \leq \sup_{f_s' \in \mathcal{F}} (\mathrm{disp}_{D_t}^{(\varrho)}(f_s', f_s)) - \inf_{f_s'' \in \mathcal{F}} (\mathrm{disp}_{D_s}^{(\varrho)}(f_s'', f_s)),$$

$$\tag{6}$$

where both $\widetilde{f}'_s$ and $\widetilde{f}''_s$ are trained by $\widetilde{f}_s$ using $D_t$ and $\tilde{D}_s$. And both $f'$ and $f''$ are trained $f_s$ using $D_t$ and $D_s$.

Proof. Please refer to Appendix A.2.

In Theorem 2, we compare the values of MDD ($d_{f,F}^{(\varrho)}(D_s, D_t)$ and $d_{f,F}^{(\varrho)}(\tilde{D}_s, D_t)$) after they are both expanded to two terms. And we obtain that after converting $D_s$ to $\tilde{D}_s$, the latter will be less than the former. In combination with Theorem 1, we conclude that in partial-set scenarios, removing categories that only exist in the source domain can reduce the differences between the source and target domains, thereby facilitating model transfer. However, it is not possible to directly filter the samples from the source domain and retrain. Instead, we leverage the model's forgetting mechanism to let the source model forget classes unique to the source domain as much as possible while retaining knowledge of the target domain classes. This effectively addresses the negative transfer problem caused by the significant differences between the source and target domains in the partial-set scenario.

### 3.3 TARGET DOMAIN PSEUDO-LABEL FILTERING

We employ a simple yet effective method to adapt the model to the target domain, specifically through the use of pseudo-labels from the target domain for self-supervised learning, which helps to demonstrate the effectiveness of our forgetting mechanism. The acquisition of pseudo-labels sample set on $D_t$ is as follows::

$$D_{tw} = \{(x_t, h_s(x_t)) | h_s(x_t) \in C_t\}, \tag{7}$$

$$D_{tK} = \{(x_t, h_s(x_t)) | \text{rank}(f_s(x_t)) \leq \text{K}\}, \tag{8}$$

where $D_{tw}$ represents the set corresponding to $C_t$ in the model's prediction of the target domain data, $D_{tK}$ is obtained by further filtering the scores predicted by the model based on $D_{tw}$, and $f_s(x_t)$ is the score of $h_s(x_t)$. We rank the predicted samples of each category from highest to lowest score and select only the top $K$ samples with the highest scores for each category. In the subsequent stages, we use $D_{tK}$ for self-supervised training of the model, which can reduce pseudo-label noise.

### 3.4 NOISE SAMPLE GENERATION

We can easily obtain the category set $C_f$ that only exists in the source domain through $C_f = C_s - C_t$. We know that the source domain model $h_s$ is obtained by minimizing the loss of all classes in $C_s$. Hence, inspired by Li et al. (2024); Tarun et al. (2023), we learn the anti-samples of the class set $C_f$ and use these anti-samples to update the model, making the model to forget about $C_f$ and thus prompt the source domain model to get back to a state where it has never seen $C_f$. We consider randomly initializing a batch of noise matrices $N$ through a normal distribution $\mathcal{N}(0, 1)$, and optimizing the noise matrix $N$ as our anti-samples by solving the following optimization problem:

$$\underset{N}{\arg\min} \, E_{(\theta)}[-L(h_s, y) + \lambda ||W_{noise}||], \tag{9}$$

where $L(\cdot, \cdot)$ is the cross-entropy loss function with $L_2$ normalization, $y \in C_f$ represents the category that needs to be forgotten. $W_{noise}$ is the parameter of the noise matrix $N$, which can also be interpreted as the pixel value of the image. $\lambda$ is the balance parameter, and in the experiment we set $\lambda = 0.1$. The former term is optimized to obtain the noise matrix $N$, while the latter term can prevent the value of $W_{noise}$ from becoming too large. The noise matrix $N$ obtained from training can be regarded as the noise sample image $x_N$ corresponding to the category that needs to be forgotten. We define the noise sample set $D_{noise} = \{(x_N, y_N) | y_N \in C_f\}$.

### 3.5 THE FORGETTING STAGE AND THE ADAPTATION STAGE

After obtaining the required sample set, we demonstrate how to train the model in this section. The noise sample set $D_{noise} = \{(x_N, y_N) | y_N \in C_f\}$ lead the model to forget $C_f$. However, forgetting will inevitably lead to the forgetting of $C_t$, which is undesirable. Therefore, we hope to constrain the model's changes on $C_t$ during the forgetting stage. The loss function we used in the forgetting stage is:

$$L_f = L_{ce}(y_t, h_s(x_t)) + \alpha L_{ce}(y_N, h_s(x_N)), \tag{10}$$

where $L_{ce}$ represents cross entropy loss, $x_t \in D_{tK}$, $x_N \in D_{noise}$, $\alpha$ represents the balance parameter.

To minimize the impact of the model's memory of $C_t$ during the forgetting stage as much as possible, we choose a smaller number of training iterations. By minimizing the loss $L_f$, we obtain the forgetting model $h_f$ from the pre-trained model $h_s$. Then, in the adaptation stage, we also need to adapt the model to the target domain. Specifically, we train $h_f$ on the obtained $D_{tK}$ using the following loss:

$$L_a = L_{ce}(x_t, h_f(x_t)), \tag{11}$$

where $x_t \in D_{tK}$.

### 3.6 ITERATIVE TRAINING

In this section, we introduce the iterative training of the model. Due to the domain shift between the source and target domains, we use multiple iterative steps to continuously update $D_{tK}$ and $D_{noise}$, and further reduce the negative transfer problem of partial-set, improving the accuracy of the target domain model $h_t$. Thus, we achieve better results than single-step training. At each iteration, we update $D_{tK}$ and $D_{noise}$ through Eqs.8 and 9, and then update the target domain model $h_t$ through Eqs.10 and 11. In Alg.1, we summarize the entire training process of our Machine Unlearning Framework.

---

**Algorithm 1** Algorithm of Machine Unlearning Framework.

---

**Input**: Target dataset $D_t$, Target domain category set $C_t$, Source domain category set $C_s$, Source pre-training model $h_s$ and its scoring function $f_s(x)$.
**Parameter**:E, F, A, $\alpha$, $\lambda$, K.
**Output**: Target model $h_t$.

 1: **for** ep = 1 to E **do**
 2:     Obtain $D_{tK}$ using Eqs.7 and 8.
 3:     $C_f = C_s - C_t$.
 4:     Obtain $D_{noise}$ using Eq.9.
 5:     **for** epochs = 1 to F **do**
 6:         Update the model from $h_s$ to $h_f$ using Eq.10.
 7:     **end for**
 8:     **for** epochs = 1 to A **do**
 9:         Update the model from $h_f$ to $h_t$ using Eq.11.
10:     **end for**
11:     $h_s$=$h_t$.
12: **end for**
13: **return** $h_t$.

---

## 4 EXPERIMENTS

### 4.1 SETUP

**Datasets.** We evaluate our method on two widely used SFUDA datasets: **Office-31**(Saenko et al., 2010) and **Office-Home**(Venkateswara et al., 2017). **Office-31** is a standard DA benchmark which contains three domains (Amazon (A), DSLR (D), and Webcam (W)) and each domain consists of 31 classes. **Office-Home** is a challenging medium-sized benchmark, which consists of four distinct domains (Artistic (Ar), Clipart (Cl), Product (Pr), and Real-World (Rw)), and each domain consists of 65 classes.

**Baselines.** As a pluggable framework, we compare the accuracy of ResNet-50(He et al., 2016), TPDS(Tang et al., 2024), Sticker(Kundu et al., 2022), CAiDA(Dong et al., 2021) and SHOT(Liang et al., 2020) with the addition of our framework. They are all well-known methods in the field of SFUDA. Specifically, CAiDA is a method for knowledge adaptation from multiple source domains to an unlabeled target domain.

**Implementation Details.** We employ ResNet-50 as the backbone for **Office-31** and **Office-Home**. For **Office-31**, we train 50 epochs to obtain a pre-trained model. For **Office-Home**, we train 100 epochs to obtain a pre-trained model. Most hyperparameters of our method do not require heavy tuning. We set 5 epochs during the forgetting stage and 60 epochs during the adaptation stage. We set $K = 7$ for each category of $C_t$ and generate 32 noise samples for each category of $C_f$. For **Office-31**, we set $\alpha = 5$ and iteratively train our method 5 times. For **Office-Home**, we set $\alpha = 1$ and only fully implemente our method once. All experiments are conducted with PyTorch on NVIDIA 3070 GPUs.

To simulate complex partial-set scenarios in reality, we have set multiple different target domain categories ($C_t$) to verify the effectiveness of our method. We observe that different methods consistently underperform on the same set of categories. Besides, if it can be verified that our method improves model accuracy for both the categories with the worst and the best performance, we can conclude that our approach is effective. Therefore, on **Office-31**, we divide the target domain categories into two parts: the 6 worst accuracy classes ($C_{t6}$) and the 25 highest accuracy classes ($C_{t25}$). On **Office-Home**, we divide the target domain categories into three parts: the 5 worst accuracy classes ($C_{t5}$), the 15 worst accuracy classes ($C_{t15}$), and the 50 highest accuracy classes ($C_{t50}$). We use one or several domains of the dataset as the source domain, and then one of the remaining domains as the target domain. Specifically, we find that on **Office-31**, pre-trained models can achieve excellent results when the target domain is Webcam or DSLR. Therefore, our target domain on **Office-31** is fixed to Amazon.

## 4.2 Results of Office-31

We obtain different pre-trained models under single source domain adaptation (SSDA) and multi-source domain adaptation (MSDA) settings. The columns named $D \rightarrow A$ and $W \rightarrow A$ in Table 1 presents the results of our method under SSDA settings. We observe that for different pre-trained models and in two different target domain category settings, there is a significant improvements after adding our method. The columns named $\rightarrow A$ in Table 1 shows the results of our method under MSDA settings. Our method still demonstrates excellent performance, especially the initial accuracy of the CAiDA model for $C_{t25}$ has reached a satisfactory 85.71%, and adding our method can still achieve a significant improvement of 3.17%. In addition, we also test the effectiveness of our method in solving negative transfer problems. Specifically, after inputting the target domain images into the model, we count the number of negative transitions for all predicted images labeled as category $C_f$. We also observe the negative transitions are more likely to occur in categories where methods have poor performance. It can be observed that after implementing our method, the number of negative transfer samples in these categories has decreased significantly. For example, for $C_{t6}$, the negative transfer sample of Resnet50 decreases from 312 to 23, which greatly improves the model's accuracy.

Table 1: Classification accuracy (%) on **Office-31** dataset. n/t represents the number of negative transfer samples over the total samples under the MSDA settings.

| Method | $C_t$ | Add ours | D→A | W→A | Avg. | →A | n/t |
|---|---|---|---|---|---|---|---|
| Resnet50 | $C_{t6}$ | × | 30.26 | 30.30 | 30.28 | 31.45 | 312/585 |
| | | ✓ | 62.22 | 68.03 | 65.13 | 72.99 | 23/585 |
| | $C_{t25}$ | × | 68.73 | 71.95 | 70.34 | 73.52 | 109/2232 |
| | | ✓ | 86.78 | 86.51 | 86.65 | 86.33 | 0/2232 |
| SHOT | $C_{t6}$ | × | 34.53 | 31.62 | 33.08 | 34.87 | 280/585 |
| | | ✓ | 53.84 | 38.11 | 45.98 | 57.09 | 59/585 |
| | $C_{t25}$ | × | 75.62 | 72.40 | 74.01 | 85.71 | 39/2232 |
| | | ✓ | 86.87 | 87.54 | 87.21 | 88.88 | 12/2232 |
| Sticker | $C_{t6}$ | × | 36.58 | 34.19 | 35.39 | 33.84 | 289/585 |
| | | ✓ | 44.44 | 44.44 | 44.44 | 59.66 | 65/585 |
| | $C_{t25}$ | × | 78.22 | 80.29 | 79.26 | 84.41 | 47/2232 |
| | | ✓ | 84.13 | 87.90 | 86.02 | 87.23 | 25/2232 |

## 4.3 RESULTS OF OFFICE-HOME

Table 2 shows the effectiveness of our method under MSDA settings. We find that for $C_{t5}$, the accuracy of different pre-trained models is between 30% and 40%, while after incorporating our method, the accuracy is greater than 40%. And for $C_{t50}$, although pre-trained models can achieve high original average accuracy, our method can still effectively improve the performance of the model. Table 3 shows the performance of our method under SSDA settings, where we train the source domain model using Resnet50 through Art. It is obvious to see that after adding our method, the number of negative transfer samples in different target domain settings of the model is greatly reduced, and the accuracy of the model has been improved.

Table 2: Classification accuracy (%) on **Office-Home** dataset under MSDA settings.

| Method | $C_t$ | Add ours | →Ar | →Cl | →Pr | →Rw | Avg. |
|---|---|---|---|---|---|---|---|
| Resnet50 | $C_{t5}$ | × | 27.71 | 23.21 | 36.36 | 47.55 | 33.71 |
| | | ✓ | 35.54 | 37.20 | 54.55 | 54.90 | 45.55 |
| | $C_{t15}$ | × | 37.91 | 28.63 | 50.91 | 58.26 | 43.93 |
| | | ✓ | 48.61 | 41.84 | 61.68 | 63.74 | 53.97 |
| | $C_{t50}$ | × | 74.36 | 61.88 | 86.93 | 88.08 | 77.81 |
| | | ✓ | 78.62 | 65.63 | 87.80 | 88.64 | 80.42 |
| Sticker | $C_{t5}$ | × | 14.71 | 12.57 | 64.67 | 52.22 | 36.04 |
| | | ✓ | 22.55 | 19.43 | 66.85 | 53.24 | 40.52 |
| | $C_{t15}$ | × | 38.06 | 40.07 | 81.34 | 62.10 | 55.39 |
| | | ✓ | 44.44 | 43.73 | 84.70 | 65.31 | 59.55 |
| | $C_{t50}$ | × | 82.44 | 69.32 | 84.56 | 89.42 | 81.43 |
| | | ✓ | 83.33 | 69.96 | 85.49 | 89.75 | 82.13 |
| CAiDA | $C_{t5}$ | × | 25.16 | 15.17 | 52.52 | 39.47 | 33.08 |
| | | ✓ | 41.72 | 19.31 | 64.64 | 47.37 | 43.26 |
| | $C_{t15}$ | × | 41.00 | 35.28 | 67.43 | 55.37 | 49.77 |
| | | ✓ | 52.60 | 39.41 | 71.20 | 59.73 | 55.74 |
| | $C_{t50}$ | × | 76.36 | 62.13 | 86.65 | 86.95 | 78.02 |
| | | ✓ | 79.10 | 63.47 | 87.60 | 87.87 | 79.51 |

Table 3: Classification accuracy (%) and number of negative transfer samples (n) on **Office-Home** dataset under SSDA settings.

| Method | $C_t$ | Add ours | Ar→Cl | n | Ar→Pr | n | Ar→Rw | n |
|---|---|---|---|---|---|---|---|---|
| Resnet50 | $C_{t5}$ | × | 17.56 | 274 | 17.91 | 293 | 41.26 | 151 |
| | | ✓ | 47.92 | 106 | 41.05 | 200 | 63.29 | 78 |
| | $C_{t15}$ | × | 23.42 | 652 | 31.97 | 476 | 52.58 | 329 |
| | | ✓ | 56.06 | 208 | 60.66 | 154 | 58.39 | 129 |
| | $C_{t50}$ | × | 51.43 | 236 | 74.16 | 168 | 79.85 | 128 |
| | | ✓ | 61.14 | 83 | 86.03 | 27 | 88.20 | 19 |
| TPDS | $C_{t5}$ | × | 42.15 | 193 | 27.82 | 223 | 58.39 | 104 |
| | | ✓ | 52.98 | 123 | 45.45 | 111 | 59.79 | 72 |
| | $C_{t15}$ | × | 43.34 | 504 | 50.23 | 302 | 62.09 | 241 |
| | | ✓ | 52.55 | 296 | 59.41 | 156 | 69.84 | 144 |
| | $C_{t50}$ | × | 63.87 | 411 | 85.61 | 237 | 87.19 | 129 |
| | | ✓ | 68.51 | 301 | 89.43 | 114 | 88.76 | 68 |

## 4.4 ABLATION STUDY

**Component Analysis.** To investigate the necessity of our module and test the time consumption of our method, we conduct ablation experiments as shown in Table 4. And our experiments on **Office-Home** are included in Appendix B. **Resnet50+only forget** represents only using our forgetting stage, **Resnet50+only adapt** represents only using our adaptation stage, **Resnet50+ours** represents using the entire Machine Unlearning framework and the iteration number is one, **Resnet50+ours * 3** represents using the entire Machine Unlearning framework and the iteration

number is three, and so on. It is evident that our method takes very short time, since the generation of $D_{tK}$ and $D_{noise}$ in each iteration is efficient. And the first iteration of our method resulted in larger improvement than only using the forgetting stage and only using the adaptation stage, and the subsequent iterative process also has a positive effect on the model, gradually reaching a stable state. It can be observed that the number of negative transfer samples can gradually decrease through iterative training.

**Analysis of $K$.** Figure 3 shows the parameter experiment on **Office-31**, with the purpose of investigating the impact of different $K$ values on the performance of our method. We hope to obtain as many and high-quality $D_{tK}$ as possible, but usually there is a trade-off between quantity and quality. Under the condition of $C_{t6}$ category setting and $K = 4$, the accuracy of **Resnet50+ours * 5** is only 63.25%, and there are 122 negative transfer samples. When $K$ is set to 6, the accuracy is improved by 8.89% and the number of negative transfer samples is reduced by 99. In addition, we also observe that when $K = 8$, the number of negative transfers increases, which is due to the introduction of pseudo label noise. Considering both model accuracy and the number of negative transfer samples, we ultimately set $K = 7$.

Table 4: Ablation study on **Office-31** dataset under MSDA settings.(Resnet50, $C_{t6}$)

| Method | time/s | $\rightarrow$A | n/t |
|---|---|---|---|
| Resnet50 | 1420 | 31.45 | 312/585 |
| Resnet50+only forget | 1552 | 34.70 | 0/585 |
| Resnet50+only adapt | 1488 | 56.92 | 151/585 |
| Resnet50+ours | 1586 | 56.61 | 121/585 |
| Resnet50+ours*3 | 1906 | 71.28 | 32/585 |
| Resnet50+ours*5 | 2236 | 72.99 | 23/585 |
| Resnet50+ours*7 | 2566 | 73.11 | 22/585 |

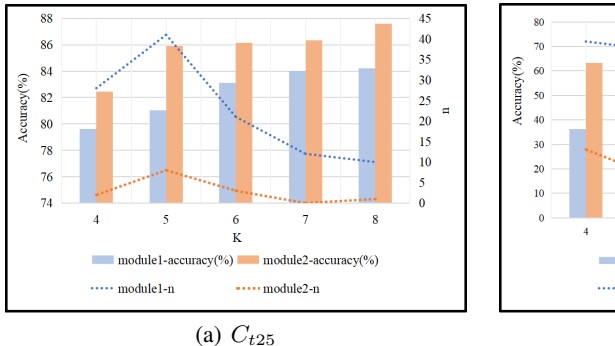 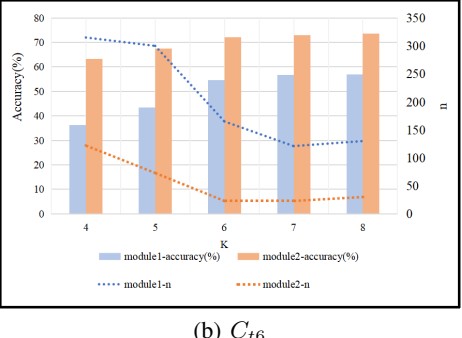

(a) $C_{t25}$          (b) $C_{t6}$

Figure 3: Performance sensitivity of parameter $K$ on **Office-31** dataset, where module1 represents **Resnet50+ours** and module2 represents **Resnet50+ours*5**, $n$ represents the number of negative transfer samples.

## 5 CONCLUSION

In this paper, we aim to address the negative transfer problem in the partial-set scenario of SFUDA, to achieve satisfactory performance of the model in the target domain. We have demonstrated that reducing the differences between source and target domains in a partial-set scenario is beneficial. Based on this, we propose a Machine Unlearning framework to solve the negative transfer problem, and experiments show that our method significantly reduces the number of negative transfer samples, effectively alleviating the issue of negative transfer in partial-set, thereby improving the accuracy of the model. In addition, as a simple and effective pluggable method, our method is suitable for various deep networks.

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

# A   PROOFS

## A.1   PROOF OF THEOREM 1

**Theorem 1.** For the target domain model $h_t$,

$$err_{D_t}(h_t) \leq err_{D_s}^{(\varrho)}(f_t) + \sup_{f' \in \mathcal{F}}(disp_{D_t}^{(\varrho)}(f', f_s)) - \inf_{f' \in \mathcal{F}}(disp_{D_s}^{(\varrho)}(f'', f_s)) + \lambda, \qquad (12)$$

where $\lambda = \min_{f^* \in \mathcal{F}}\{err_{D_s}^{(\varrho)}(f^*) + err_{D_t}^{(\varrho)}(f^*)\}$ is independent of $h_t$. Both $f'$ and $f''$ are trained by $f_t$ using $D_t$ and $D_s$.

**Lemma 1.** For the target domain model Zhang et al. (2019),

$$err_{D_t}(h_t) \leq err_{D_s}^{(\varrho)}(f_t) + d_{f,F}^{(\varrho)}(D_s, D_t) + \lambda, \qquad (13)$$

where $\lambda = \min_{f^* \in \mathcal{F}}\{err_{D_s}^{(\varrho)}(f^*) + err_{D_t}^{(\varrho)}(f^*)\}$ is independent of $h_t$.

Proof. In a hypothetical space $F$, we have $\hat{f}$ satisfies

$$d_{f,F}^{(\varrho)}(D_s, D_t) \triangleq \sup_{\hat{f} \in F}(disp_{D_t}^{(\varrho)}(\hat{f}, f) - disp_{D_s}^{(\varrho)}(\hat{f}, f)), \qquad (14)$$

Then we have

$$disp_{D_t}^{(\varrho)}(\hat{f}, f) \leq \sup_{f' \in F}(disp_{D_t}^{(\varrho)}(f', f_s)), \qquad (15)$$

$$\inf_{f'' \in F}(disp_{D_s}^{(\varrho)}(f'', f_s)) \leq disp_{D_s}^{(\varrho)}(\hat{f}, f). \qquad (16)$$

Therefore, we have

$$d_{f,F}^{(\varrho)}(D_s, D_t) \leq \sup_{f' \in \mathcal{F}}(disp_{D_t}^{(\varrho)}(f', f_s)) - \inf_{f'' \in F}(disp_{D_s}^{(\varrho)}(f'', f_s)). \qquad (17)$$

Based on **Lemma 1**, we have

$$\begin{aligned} err_{D_t}(h_t) \leq err_{D_s}^{(\varrho)}(f_t) + \sup_{f^* \in F}(disp_{D_t}^{(\varrho)}(f^*, f_s)) \\ - \inf_{f^* \in F}(disp_{D_s}^{(\varrho)}(f^*, f_s)) + \lambda. \end{aligned} \qquad (18)$$

## A.2   PROOF OF THEOREM 2

**Theorem 2.** In the same hypothesis space $F$, $h_s$ is trained on $D_s$, and $\tilde{h}_s$ is trained on $\tilde{D}_s$,

$$\sup_{\widetilde{f}_s' \in \mathcal{F}}(\mathrm{disp}_{D_t}^{(\varrho)}(\widetilde{f}_s', \widetilde{f}_s)) - \inf_{\widetilde{f}_s'' \in \mathcal{F}}(\mathrm{disp}_{\tilde{D}_s}^{(\varrho)}(\widetilde{f}_s'', \widetilde{f}_s)) \leq \sup_{f_s' \in \mathcal{F}}(\mathrm{disp}_{D_t}^{(\varrho)}(f_s', f_s)) - \inf_{f_s'' \in \mathcal{F}}(\mathrm{disp}_{D_s}^{(\varrho)}(f_s'', f_s)), \qquad (19)$$

where both $\widetilde{f}_s'$ and $\widetilde{f}_s''$ are trained by $\widetilde{f}_s$ using $D_t$ and $\tilde{D}_s$. And both $f'$ and $f''$ are trained $f_s$ using $D_t$ and $D_s$.

Proof. For any $x \in D_s$, we have

$$f_s''(x, h_s(x)) - \max_{y' \neq h_s(x)} f_s''(x, y')$$

$$\leq f_s(x, h_s(x)) - \max_{y' \neq h_s(x)} f_s(x, y'). \qquad (20)$$

According to **Definition 2**, we have

$$\begin{aligned} disp_{D_s}^{(\varrho)}(f_s'', f_s) &\triangleq \mathbb{E}_{x \sim D_s}\Phi_\varrho\left(\varrho_{f_{s''}}(x, h_s(x))\right) \\ &\geq \mathbb{E}_{x \sim D_s}\Phi_\varrho\left(\varrho_{f_s}(x, h_s(x))\right) \\ &\geq 0. \end{aligned} \qquad (21)$$

When $f_s'' = f_s$ and $\varrho$ is a sufficiently small positive number, we have $\inf_{f_s'' \in \mathcal{F}}(disp_{D_s}^{(\varrho)}(f_s'', f_s)) = 0$. Similarly, when $\widetilde{f}_s'' = \widetilde{f}_s$ and $\varrho$ is a sufficiently small positive number, we have $\inf_{\widetilde{f}_s'' \in \mathcal{F}}(disp_{\widetilde{D}_s}^{(\varrho)}(\widetilde{f}_s'', \widetilde{f}_s)) = 0$.

Let $D_{t1} = \{(x_t, h_s(x_t)) | h_s(x_t) \in C_f\}$. For any $x_t \in D_{t1}$, we have

$$f_s'(x_t, h_s(x)) < \max_{y' \neq h_s(x)} f_s'(x_t, y'). \tag{22}$$

Then, for $D_{t1}$, we have

$$\varrho_{f_s'}(x_t, h_s(x_t)) < 0. \tag{23}$$

Then, we can obtain $\mathrm{G}_\varrho(\,\varrho_{f_s'}(x_t, h_s(x_t))\,) = 1$. So we have

$$disp_{D_{t1}}^{(\varrho)}(\tilde{f}_s', \tilde{f}_s) < disp_{D_{t1}}^{(\varrho)}(f_s', f_s) = 1. \tag{24}$$

Let $D_{t2} = D_t - D_{t1}$. For $D_{t2}$, we can reasonably assume that we have $h_s(x_t) = \tilde{h}_s(x_t)$.

Then, we have

$$disp_{D_{t2}}^{(\varrho)}(f_s', f_s) = disp_{D_{t2}}^{(\varrho)}(\widetilde{f}_s', \widetilde{f}_s). \tag{25}$$

Therefore, we have

$$\sup_{\widetilde{f}_s' \in \mathcal{F}}(\mathrm{disp}_{D_t}^{(\varrho)}(\widetilde{f}_s', \widetilde{f}_s)) \leq \sup_{f_s' \in \mathcal{F}}(\mathrm{disp}_{D_t}^{(\varrho)}(f_s', f_s)). \tag{26}$$

Then we have

$$\sup_{\widetilde{f}_s' \in \mathcal{F}}(\mathrm{disp}_{D_t}^{(\varrho)}(\widetilde{f}_s', \widetilde{f}_s)) - \inf_{\widetilde{f}_s'' \in \mathcal{F}}(\mathrm{disp}_{\tilde{D}_s}^{(\varrho)}(\widetilde{f}_s'', \widetilde{f}_s))$$

$$\leq \sup_{f_s' \in \mathcal{F}}(\mathrm{disp}_{D_t}^{(\varrho)}(f_s', f_s)) - \inf_{f_s'' \in \mathcal{F}}(\mathrm{disp}_{D_s}^{(\varrho)}(f_s'', f_s)). \tag{27}$$

## B  ADDITIONAL EXPERIMENTS

Table 5: Ablation study about training time on **Office-Home** dataset under SSDA settings.(Ar→Cl)

| Method | time/s | | |
|---|---|---|---|
|  | $C_{t5}$ | $C_{t15}$ | $C_{t50}$ |
| Resnet50 |  | 1550 |  |
| TPDS |  | 3137 |  |
| TPDS+only forget | 3860 | 3765 | 3371 |
| TPDS+only adapt | 3183 | 3230 | 3364 |
| TPDS+ours | 3906 | 3858 | 3598 |

We also evaluate the training time of our method on **Office-Home**, and the results are shown in Table 5. It can be observed that our adaptation stage takes a very short time, while the forgetting stage takes time that is proportional to the number of categories to be forgotten. However, even in the setting where the target domain is $C_{t5}$ (requiring forget 50 categories), the time taken by our algorithm is shorter than TPDS, which takes 1587 seconds for training.

