# OpenReview forum: "Machine Unlearning For Alleviating Negative Transfer In Partial-Set Source-Free Unsupervised Domain Adaptation"
_ICLR.cc/2025/Conference — ICLR 2025 Conference Withdrawn Submission_

### Official Review · Reviewer_UDZN · 2024-10-21

**Soundness:** 3
**Presentation:** 2
**Contribution:** 2
**Rating:** 5
**Confidence:** 3

**Summary:**

This paper tackles Source-Free Unsupervised Domain Adaptation (SFUDA) in a partial-set scenario, where the source label space subsumes the target one, risking negative transfer. The authors propose a Machine Unlearning Framework to address this issue by first generating noise samples for source-only categories and pseudo-labeled samples from the target domain. In the forgetting stage, the model is trained to forget irrelevant source categories, while in the adaptation stage, it undergoes self-supervised learning on target data. The framework is simple, adaptable to various models, and improves performance across different target category settings. Experimental results show its effectiveness in reducing negative transfer and enhancing accuracy.

**Strengths:**

*	The proposed method is effective for SFUDA.
*	The paper is well-written and easy to follow.

**Weaknesses:**

*	Experiments on large-scale datasets, such as DomainNet, are missing. Including comparisons on such datasets would further demonstrate the flexibility of the proposed method.
*	Algorithm 1 can be improved. For example, the authors could provide more details on line 3. Offering further descriptions for each step of Algorithm 1 would enhance the paper's readability.
*	The comparison methods on Tabs. 1 and 2 are out-of-date. It is recommended that the authors include more recent approaches proposed in 2023 and 2024.
*	The experimental tasks in Tabs. 1, 2, and 3 are also insufficient. For instance, in Tab. 1, the authors should add the tasks A→D, A→W, and W→D for SSDA, and →D, →W for MSDA.

**Questions:**

*	Could the authors include more visualization analyses (e.g., t-SNE, CAM, or confusion matrix)? Adding such analyses would further enhance the quality of the paper and demonstrate the effectiveness of the proposed method.
*	The font size on Fig. 2 seems a little bit small, could the authors adjust it?
*	Typo: $x_s^i \to x_t^i$ in line 189.

---

> ### Author Response · Authors · 2024-11-22
>
> Thank you very much for the great comments. Our response to the your questions are elaborated below.
>
> $\textcolor{orange}{\text{Q1:}} $ Experiments on large-scale datasets, such as DomainNet, are missing. Including comparisons on such datasets would further demonstrate the flexibility of the proposed method.
>
> $\textcolor{green}{\text{Response:}}$ Great sugestion. We have tested our method on the DomainNet-126 dataset. Similar to our other experiments, we set the target domain classes to the 30 categories with the lowest accuracy ($C_{t30}$). The experimental results are presented in the table below. We observe a significant improvement in the accuracy of our method compared to different initial models.
>
> Table: Classification accuracy (\%) on 7 domain shifts of $\textbf{DomainNet-126}$ dataset under SSDA settings. All methods use the ResNet-50 backbone. ($C_{t30}$)
> | Method      | Add ours   | S→P   | C→S   | P→C   | P→R   | R→S   | R→C   | R→P   | Avg.  |
> |-------------|------------|-------|-------|-------|-------|-------|-------|-------|-------|
> | Resnet50    | $\times$   | 19.69 | 23.10 | 29.29 | 48.55 | 25.73 | 30.37 | 32.91 | 29.95 |
> |             | $\checkmark$| 33.55| 31.37 | 32.82 | 48.20 | 26.03 | 36.85 | 33.25 | 34.58 |
> | GKD[1]         | $\times$   | 28.60 | 24.15 | 35.41 | 54.29 | 30.07 | 39.58 | 30.17 | 34.61 |
> |             | $\checkmark$| 34.40| 34.63 | 39.42 | 49.31 | 32.65 | 39.42 | 33.97 | 37.69 |
> | PLUE[2]        | $\times$   | 31.45 | 24.97 | 29.42 | 49.09 | 28.06 | 30.33 | 33.59 | 32.40 |
> |             | $\checkmark$| 34.90| 29.78 | 33.19 | 49.99 | 29.16 | 39.16 | 34.47 | 35.81 |
> | TPDS[3]        | $\times$   | 29.21 | 26.76 | 35.44 | 53.06 | 30.76 | 37.79 | 32.10 | 35.02 |
> |             | $\checkmark$| 33.03| 34.39 | 45.31 | 50.47 | 37.22 | 40.11 | 34.85 | 39.34 |
>
> [1] Model adaptation through hypothesis transfer with gradual knowledge distillation, In IROS21.
>
> [2] Guiding Pseudo-labels with Uncertainty Estimation for Source-free Unsupervised Domain Adaptation, In CVPR23.
>
> [3] Source-free domain adaptation via target prediction distribution searching, In IJCV24.
>
> $\textcolor{orange}{\text{Q2、Q6 and Q7:}} $Algorithm 1 can be improved. For example, the authors could provide more details on line 3. Offering further descriptions for each step of Algorithm 1 would enhance the paper's readability. The font size on Fig. 2 seems a little bit small, could the authors adjust it? Typo: $x^{i}_{s} ->x^{i}_{t} $  in line 189.
>
> $\textcolor{green}{\text{Response:}}$ Thank you very much for your reminder. We will correct our mistakes and update our framework diagram and pseudocode.
>
> $\textcolor{orange}{\text{Q4:}} $The experimental tasks in Tabs. 1, 2, and 3 are also insufficient. For instance, in Tab. 1, the authors should add the tasks A→D, A→W, and W→D for SSDA, and →D, →W for MSDA.
>
> $\textcolor{green}{\text{Response:}}$Thank you for your insightful suggestion. In our analysis of the Office31 dataset, we observed that when Webcam or DSLR serves as the target domain, many current SFUDA methods [4-6] achieve extremely high accuracy rates exceeding 95% or even 99%. These exceptional results indicate that negative transfer is virtually non-existent in these scenarios.
> Therefore, to provide a more meaningful evaluation of our method's effectiveness in handling negative transfer, we specifically chose to fix Amazon as the target domain in our Office31 experiments. This choice is more challenging and better demonstrates the robustness of our approach.
>
> [4] Source Data-absent Unsupervised Domain Adaptation through Hypothesis Transfer and Labeling Transfer, In T-PAMI21.
>
> [5] Confident-Anchor-Induced Multi-Source-Free Domain Adaptation, In NIPS21.
>
> [6] Concurrent Subsidiary Supervision for Unsupervised Source-Free Domain Adaptation, In ECCV22.

---

### Official Review · Reviewer_5Wgg · 2024-10-27

**Soundness:** 3
**Presentation:** 2
**Contribution:** 2
**Rating:** 5
**Confidence:** 4

**Summary:**

Aiming at the problem of partial domain adaptation, this paper proposes the method of machine unlearning. To eliminate the negative transfer effect of source-specific classes on target adaptation, experiments show the effectiveness of the method.

**Strengths:**

1. The idea of eliminating the influence of  source-specific classes by using machine unlearning is new to some extent, which provides a new insight compared with the classical thoughts of eliminating negative transfer by weighting in adversarial learning in the past.
2. Ablation study was sufficient to demonstrate the effectiveness of the method.

**Weaknesses:**

1. The frame diagram is not very intuitive, it is best to label each subgraph, and combine the label to explain the frame diagram, N(0,1) in the figure has no explanation, the reviewer needs to find a definition in the main text to understand what to do. And there are many symbols, it is not easy to understand, it is suggested to further optimize the frame diagram and interpretation.
2. Experiments were only performed on the smaller scale of Office-Home/Office-31 and not on the larger scale of ImageNet/VisDA-2017 dataset, which have been extensively covered by classical methods in the past [1][2]
3. The author claims in line 301: "We can easily obtain the category set $C_f$ that only exists in the source domain". However, the method used still relies on the predictions of a model pretrained on the source domain for filtering the target samples. This step still carries a strong source domain bias, and the target samples obtained on this basis cannot guarantee that the selected categories are exclusively from the source domain.

[1]  Hu J, Tuo H, Wang C, et al. Discriminative partial domain adversarial network[C]//Computer Vision–ECCV 2020: 16th European Conference, Glasgow, UK, August 23–28, 2020, Proceedings, Part XXVII 16. Springer International Publishing, 2020: 632-648.

[2] Liang J, Wang Y, Hu D, et al. A balanced and uncertainty-aware approach for partial domain adaptation[C]//European conference on computer vision. Cham: Springer International Publishing, 2020: 123-140.

**Questions:**

1. In the experimental part, the SFUDA method should not be compared only, but should be compared with both the classic and the latest PDA methods to show the effectiveness of the method.
2. Need more explanation on "Why $C_f$ can be accurately selected out without target labels".
3. On the premise of not dealing with negative transfer, SFUDA only needs to ensure sufficient spacing between the classification boundary between classes in the case of cross-domain to achieve partial domain adaptation. So why is it necessary to design modules specifically to eliminate the effects of negative migration?

---

> ### Author Response · Authors · 2024-11-22
>
> Thank you very much for the great comments. Our response to the your questions are elaborated below.
>
> $\textcolor{orange}{\text{Q1:}} $The frame diagram is not very intuitive, it is best to label each subgraph, and combine the label to explain the frame diagram, N(0,1) in the figure has no explanation, the reviewer needs to find a definition in the main text to understand what to do. And there are many symbols, it is not easy to understand, it is suggested to further optimize the frame diagram and interpretation.
>
> $\textcolor{green}{\text{Response:}}$We will update our framework diagram and provide more detailed captions to enhance clarity.
>
> $\textcolor{orange}{\text{Q2:}} $Experiments were only performed on the smaller scale of Office-Home/Office-31 and not on the larger scale of ImageNet/VisDA-2017 dataset, which have been extensively covered by classical methods in the past.
>
> $\textcolor{green}{\text{Response:}}$ Great sugestion. We have tested our method on the DomainNet-126 dataset. Similar to our other experiments, we set the target domain classes to the 30 categories with the lowest accuracy ($C_{t30}$). The experimental results are presented in the table below. We observe a significant improvement in the accuracy of our method compared to different initial models.
>
> Table: Classification accuracy (\%) on 7 domain shifts of $\textbf{DomainNet-126}$ dataset under SSDA settings. All methods use the ResNet-50 backbone. ($C_{t30}$)
> | Method      | Add ours   | S→P   | C→S   | P→C   | P→R   | R→S   | R→C   | R→P   | Avg.  |
> |-------------|------------|-------|-------|-------|-------|-------|-------|-------|-------|
> | Resnet50    | $\times$   | 19.69 | 23.10 | 29.29 | 48.55 | 25.73 | 30.37 | 32.91 | 29.95 |
> |             | $\checkmark$| 33.55| 31.37 | 32.82 | 48.20 | 26.03 | 36.85 | 33.25 | 34.58 |
> | GKD[1]         | $\times$   | 28.60 | 24.15 | 35.41 | 54.29 | 30.07 | 39.58 | 30.17 | 34.61 |
> |             | $\checkmark$| 34.40| 34.63 | 39.42 | 49.31 | 32.65 | 39.42 | 33.97 | 37.69 |
> | PLUE[2]        | $\times$   | 31.45 | 24.97 | 29.42 | 49.09 | 28.06 | 30.33 | 33.59 | 32.40 |
> |             | $\checkmark$| 34.90| 29.78 | 33.19 | 49.99 | 29.16 | 39.16 | 34.47 | 35.81 |
> | TPDS[3]        | $\times$   | 29.21 | 26.76 | 35.44 | 53.06 | 30.76 | 37.79 | 32.10 | 35.02 |
> |             | $\checkmark$| 33.03| 34.39 | 45.31 | 50.47 | 37.22 | 40.11 | 34.85 | 39.34 |
>
> [1] Model adaptation through hypothesis transfer with gradual knowledge distillation, In IROS21.
>
> [2] Guiding Pseudo-labels with Uncertainty Estimation for Source-free Unsupervised Domain Adaptation, In CVPR23.
>
> [3] Source-free domain adaptation via target prediction distribution searching, In IJCV24.
>
> $\textcolor{orange}{\text{Q3:}} $ The author claims in line 301: "We can easily obtain the category set $C_f$
>  that only exists in the source domain". However, the method used still relies on the predictions of a model pretrained on the source domain for filtering the target samples. This step still carries a strong source domain bias, and the target samples obtained on this basis cannot guarantee that the selected categories are exclusively from the source domain.
>
> $\textcolor{green}{\text{Response:}}$You are correct that using the source domain model to obtain pseudo-labels for the target domain can introduce source domain bias, a concern that many SFUDA methods aim to address. Our framework’s iterative process incorporates frequent forgetting steps to minimize this bias, thereby helping to reduce negative transfer occurrences.

---

> > ### Author Response · Authors · 2024-11-22
> >
> > Thank you for your valuable feedback. Our response to the your questions are elaborated below.
> >
> > $\textcolor{orange}{\text{Q1:}} $In the experimental part, the SFUDA method should not be compared only, but should be compared with both the classic and the latest PDA methods to show the effectiveness of the method.
> >
> > $\textcolor{green}{\text{Response:}}$Indeed, comparing our method to PDA highlights its effectiveness. However, it's important to note that PDA primarily relies on the availability of source domain data, whereas our approach does not utilize any source domain data at all. This makes a direct comparison somewhat unfair.
> >
> > $\textcolor{orange}{\text{Q2:}} $Need more explanation on "Why $C_f$ can be accurately selected out without target labels".
> >
> > $\textcolor{green}{\text{Response:}}$ In real-world classification tasks, the target domain class space is typically well-defined and known beforehand. In our work, we focus on finding a source domain model whose class space encompasses (rather than exactly matches) the target domain class space, which can then be adapted to better suit the target classification task through our method. Therefore, we assume both $C_s$ and $C_t$ are known. Although the target domain dataset is unlabeled, we can still determine $C_f$  through the simple relation $C_f = C_s - C_t$. This assumption aligns with practical scenarios where we know the categories we need to classify in the target domain, even though we lack labeled examples for these categories.
> >
> > $\textcolor{orange}{\text{Q3:}} $On the premise of not dealing with negative transfer, SFUDA only needs to ensure sufficient spacing between the classification boundary between classes in the case of cross-domain to achieve partial domain adaptation. So why is it necessary to design modules specifically to eliminate the effects of negative migration?
> >
> > $\textcolor{green}{\text{Response:}}$Thank you for raising this important point. While theoretically, maintaining sufficient spacing between class boundaries across domains could achieve partial domain adaptation, this becomes significantly challenging in practice, particularly in datasets with a large number of classes such as Office-Home and DomainNet. In addition, due to the inherent class mismatch caused by partial set settings and the additional variability introduced by domain shift in feature distribution, negative transfer is actually inevitable in this case. Therefore, our method specifically addresses this issue.

---

### Official Review · Reviewer_HXRy · 2024-11-03

**Soundness:** 2
**Presentation:** 3
**Contribution:** 2
**Rating:** 3
**Confidence:** 5

**Summary:**

This paper proposes a machine unlearning-based module for partial source-free unsupervised domain adaptation in image classification tasks. The module aims to “unlearn” non-target classes prior to adaptation, reducing the domain shift caused by mismatched label sets between the source and target domains. Pseudo-labeling is then applied for target domain adaptation. Experiments on two benchmark datasets show that incorporating this module enhances the performance of existing methods.

**Strengths:**

1. Innovative use of machine unlearning to address label mismatch in domain adaptation tasks.
2. Demonstrated effectiveness on the Office-Home and Office-31 datasets.

**Weaknesses:**

1. Unfair Experimental Setup: The chosen setup is overly complex and seems tailored to fit the proposed method, lacking a strong motivation. It assumes that the target label set is fully known while addressing a partial label adaptation problem in a source-free context. This setup is especially questionable given the recent advancements in vision-language models, such as CLIP, which provide alternative domain adaptation solutions.

2. As a hot-swappable module, it should ideally apply across various source-free models, not only for image classification but also for other tasks like semantic segmentation and object detection. However, only a few baseline models for image classification were chosen for evaluation.

3. The evaluation lacks breadth; a more comprehensive set of benchmarks, such as VisDA, along with various source-target combinations and different backbone architectures, should be included to thoroughly validate the method’s performance.

4. The method relies on top-K pseudo-labeling and unlearning, requiring multiple inference steps, which reduces computational efficiency.

5. The theoretical analysis focuses on the domain adaptation aspect, whereas a theoretical justification of the unlearning mechanism would be more relevant and impactful. For example, why learning data noise for these labels could results unlearning of the original class.

6. While applying unlearning to domain adaptation is interesting, the approach appears to be a straightforward combination of two existing research directions, which limits its contribution—especially within a potentially biased experimental setup.

7. The study does not include a broader range of universal setup SFDA baselines, limiting the comparison and making it difficult to gauge the module's performance against widely accepted SFDA approaches.

8. The paper has some notation and formatting inconsistencies that need correction.

**Questions:**

See the weakness part.

---

> ### Author Response · Authors · 2024-11-19
>
> Thank you very much for the great comments. Our response to the your questions are elaborated below.
>
> $\textcolor{orange}{\text{Q1:}} $Unfair Experimental Setup: The chosen setup is overly complex and seems tailored to fit the proposed method, lacking a strong motivation. It assumes that the target label set is fully known while addressing a partial label adaptation problem in a source-free context. This setup is especially questionable given the recent advancements in vision-language models, such as CLIP, which provide alternative domain adaptation solutions.
>
> $\textcolor{green}{\text{Response:}}$ Thank you for your thoughtful feedback.In our work, we are addressing a practical challenge in classification tasks where the target category space is known. Specifically, we aim to update the source domain model using the SFUDA method, but finding a source domain model trained on a source domain with an exact match to the target category space is quite difficult. Typically, source domain models are trained on large datasets, which often contain a broader category space that includes the target categories.To overcome this challenge, we propose leveraging a source pre-trained model that includes the target domain category space, even if it does not match perfectly. Our method facilitates a rapid update of this model, enabling it to adapt more effectively to the target task.
>
> $\textcolor{orange}{\text{Q2:}} $ As a hot-swappable module, it should ideally apply across various source-free models, not only for image classification but also for other tasks like semantic segmentation and object detection. However, only a few baseline models for image classification were chosen for evaluation.
>
> $\textcolor{green}{\text{Response:}}$ We would like to clarify that our theoretical proof is based on transfer error in classification tasks specifically to address the domain adaptation problem in these scenarios. Additionally, it is important to note that many domain adaptation methods are specifically designed for classification tasks [1-3]. Our experiments demonstrate that our method is applicable to a wide range of classification tasks with no source model requirement.Given that our approach is a hot-swappable method, it may not be reasonable to extend the requirement for testing on tasks such as object detection and image segmentation.
>
> [1] Source-Free Domain Adaptation with Frozen Multimodal Foundation Model, In CVPR24.
>
> [2] Source-Free Domain Adaptation via Target Prediction Distribution Searching, In IJCV23.
>
> [3] Model Adaptation through Hypothesis Transfer with Gradual Knowledge Distillation, In IROS21.
>
> $\textcolor{orange}{\text{Q4:}} $ The method relies on top-K pseudo-labeling and unlearning, requiring multiple inference steps, which reduces computational efficiency.
>
> $\textcolor{green}{\text{Response:}}$ Inference and the use of target domain pseudo labels and forgetting are the core of our method. Although it requires multiple iterations, the computational cost of our method is very small, as evidenced by our time complexity experiments.
>
> $\textcolor{orange}{\text{Q5 and Q6:}}$The theoretical analysis focuses on the domain adaptation aspect, whereas a theoretical justification of the unlearning mechanism would be more relevant and impactful. For example, why learning data noise for these labels could results unlearning of the original class. While applying unlearning to domain adaptation is interesting, the approach appears to be a straightforward combination of two existing research directions, which limits its contribution—especially within a potentially biased experimental setup.
>
> $\textcolor{green}{\text{Response:}}$To our knowledge, our approach is the first to apply machine unlearning in the context of domain adaptation. Therefore, a critical question we aim to address is whether the forgetting mechanism can positively impact domain adaptation. This is central to our theory, and we have provided a theoretical justification for the rationale behind this approach.
>
> $\textcolor{orange}{\text{Q8:}} $The paper has some notation and formatting inconsistencies that need correction.
>
> $\textcolor{green}{\text{Response:}}$Thank you for your feedback. We will pay closer attention to correcting the details related to compliance and formatting issues.

---

> > ### Author Response · Authors · 2024-11-20
> >
> > Thank you very much for the great comments. Our response to the your questions are elaborated below.
> >
> > $\textcolor{orange}{\text{Q3 and Q7:}}$ The evaluation lacks breadth; a more comprehensive set of benchmarks, such as VisDA, along with various source-target combinations and different backbone architectures, should be included to thoroughly validate the method’s performance. The study does not include a broader range of universal setup SFDA baselines, limiting the comparison and making it difficult to gauge the module's performance against widely accepted SFDA approaches.
> >
> > $\textcolor{green}{\text{Response:}}$ Great sugestion. We have tested our method on the DomainNet-126 dataset. Similar to our other experiments, we set the target domain classes to the 30 categories with the lowest accuracy ($C_{t30}$). The experimental results are presented in the table below. We observe a significant improvement in the accuracy of our method compared to different initial models.
> >
> > Table: Classification accuracy (\%) on 7 domain shifts of $\textbf{DomainNet-126}$ dataset under SSDA settings. All methods use the ResNet-50 backbone. ($C_{t30}$)
> > | Method      | Add ours   | S→P   | C→S   | P→C   | P→R   | R→S   | R→C   | R→P   | Avg.  |
> > |-------------|------------|-------|-------|-------|-------|-------|-------|-------|-------|
> > | Resnet50    | $\times$   | 19.69 | 23.10 | 29.29 | 48.55 | 25.73 | 30.37 | 32.91 | 29.95 |
> > |             | $\checkmark$| 33.55| 31.37 | 32.82 | 48.20 | 26.03 | 36.85 | 33.25 | 34.58 |
> > | GKD[1]         | $\times$   | 28.60 | 24.15 | 35.41 | 54.29 | 30.07 | 39.58 | 30.17 | 34.61 |
> > |             | $\checkmark$| 34.40| 34.63 | 39.42 | 49.31 | 32.65 | 39.42 | 33.97 | 37.69 |
> > | PLUE[2]        | $\times$   | 31.45 | 24.97 | 29.42 | 49.09 | 28.06 | 30.33 | 33.59 | 32.40 |
> > |             | $\checkmark$| 34.90| 29.78 | 33.19 | 49.99 | 29.16 | 39.16 | 34.47 | 35.81 |
> > | TPDS[3]        | $\times$   | 29.21 | 26.76 | 35.44 | 53.06 | 30.76 | 37.79 | 32.10 | 35.02 |
> > |             | $\checkmark$| 33.03| 34.39 | 45.31 | 50.47 | 37.22 | 40.11 | 34.85 | 39.34 |
> >
> > [1] Model adaptation through hypothesis transfer with gradual knowledge distillation, In IROS21.
> >
> > [2] Guiding Pseudo-labels with Uncertainty Estimation for Source-free Unsupervised Domain Adaptation, In CVPR23.
> >
> > [3] Source-free domain adaptation via target prediction distribution searching, In IJCV24.

---

### Official Review · Reviewer_tB5L · 2024-11-07

**Soundness:** 2
**Presentation:** 2
**Contribution:** 2
**Rating:** 3
**Confidence:** 4

**Summary:**

This paper addresses the Partial-Set Source-Free Unsupervised Domain Adaptation (SFUDA) problem by proposing a novel pluggable framework, termed the Machine Unlearning Framework, designed to mitigate the negative transfer issue in partial-set SFUDA. The framework generates noise samples from the source’s private classes, enabling the model to unlearn and forget the information specific to these private classes. Additionally, it employs self-supervised training using pseudo-labeled target data to optimize the model. The effectiveness of the proposed method is evaluated on two benchmark datasets.

**Strengths:**

- This paper induces a new aspect to solve the partial-set source-free domain adaptation problem.
- It induces a method based on the guide of negative transfer problems.

**Weaknesses:**

- Since the target domain is unlabeled, how do you ensure that the source pre-trained model can accurately identify the target class space $C_t$​?
- How do you substantiate the claim that "the model does not require high-performance hardware," given that your method is described as a pluggable framework?
- In your abstract, you state that $h_s$ is pre-trained with one source domain. Why, then, does your experiment involve a model pre-trained on multiple source domains?
- The benchmarks used are insufficient to demonstrate the efficiency of the proposed method. The authors should consider using DomainNet and VisDA as additional benchmarks.

**Questions:**

See weaknesses

---

> ### Author Response · Authors · 2024-11-19
>
> Thank you very much for the great comments. Our response to the your questions are elaborated below.
>
> $\textcolor{orange}{\text{Q1:}} $ Since the target domain is unlabeled, how do you ensure that the source pre-trained model can accurately identify the target class space $C_t$ ？
>
> $\textcolor{green}{\text{Response:}}$ Similar to the settings in [1-2], our setting is that the target class space is a subset of the source class space , and we establish that the target class space is known. The source pre-trained model can achieve a certain level of accuracy on the target domain. Through our two stages of forgetting and adaptation, we can reduce the negative transfer phenomenon in the model's predictions for the target domain, thereby enhancing the model's ability to accurately identify the classes in the target class space.
>
> [1] Selective Partial Domain Adaptation, In BMVC22.
>
> [2] Partial Domain Adaptation without Domain Alignment, In IEEE TPAMI22.
>
> $\textcolor{orange}{\text{Q2:}} $ How do you substantiate the claim that "the model does not require high-performance hardware," given that your method is described as a pluggable framework?
>
> $\textcolor{green}{\text{Response:}}$ When we mention "low hardware requirements," we are actually referring to the low time complexity of our method. In the SFUDA setting, since we can directly access the source model, and both our forgetting and adaptation phases require relatively small amounts of data and training epochs, model updates can be completed efficiently without extensive training time. Our time complexity experiments support this - for example, while the TPDS method requires 3,137 seconds, our approach only needs an average of 650 seconds to complete. This demonstrates that our method achieves much faster adaptation while maintaining effectiveness.
>
> $\textcolor{orange}{\text{Q3:}} $ In your abstract, you state that $h_s$ is pre-trained with one source domain. Why, then, does your experiment involve a model pre-trained on multiple source domains?
>
> $\textcolor{green}{\text{Response:}}$ To clarify in the abstract, in our method, the source model is trained on the source domain data. The source domain can actually be a union of multiple domains, and incorporating models trained on multiple source domains helps enrich the diversity of our experiments. This setting of using multiple source domains is also common in experiments across many papers in the field. Including this would make our experimental evaluation more comprehensive and align with established practices in the literature.

---

> > ### Author Response · Authors · 2024-11-20
> >
> > Thank you very much for the great comments. Our response to the your questions are elaborated below.
> >
> > $\textcolor{orange}{\text{Q4:}} $ The benchmarks used are insufficient to demonstrate the efficiency of the proposed method. The authors should consider using DomainNet and VisDA as additional benchmarks.
> >
> > $\textcolor{green}{\text{Response:}}$ Great sugestion. We have tested our method on the DomainNet-126 dataset. Similar to our other experiments, we set the target domain classes to the 30 categories with the lowest accuracy ($C_{t30}$). The experimental results are presented in the table below. We observe a significant improvement in the accuracy of our method compared to different initial models.
> >
> > Table: Classification accuracy (\%) on 7 domain shifts of $\textbf{DomainNet-126}$ dataset under SSDA settings. All methods use the ResNet-50 backbone. ($C_{t30}$)
> > | Method      | Add ours   | S→P   | C→S   | P→C   | P→R   | R→S   | R→C   | R→P   | Avg.  |
> > |-------------|------------|-------|-------|-------|-------|-------|-------|-------|-------|
> > | Resnet50    | $\times$   | 19.69 | 23.10 | 29.29 | 48.55 | 25.73 | 30.37 | 32.91 | 29.95 |
> > |             | $\checkmark$| 33.55| 31.37 | 32.82 | 48.20 | 26.03 | 36.85 | 33.25 | 34.58 |
> > | GKD[1]         | $\times$   | 28.60 | 24.15 | 35.41 | 54.29 | 30.07 | 39.58 | 30.17 | 34.61 |
> > |             | $\checkmark$| 34.40| 34.63 | 39.42 | 49.31 | 32.65 | 39.42 | 33.97 | 37.69 |
> > | PLUE[2]        | $\times$   | 31.45 | 24.97 | 29.42 | 49.09 | 28.06 | 30.33 | 33.59 | 32.40 |
> > |             | $\checkmark$| 34.90| 29.78 | 33.19 | 49.99 | 29.16 | 39.16 | 34.47 | 35.81 |
> > | TPDS[3]        | $\times$   | 29.21 | 26.76 | 35.44 | 53.06 | 30.76 | 37.79 | 32.10 | 35.02 |
> > |             | $\checkmark$| 33.03| 34.39 | 45.31 | 50.47 | 37.22 | 40.11 | 34.85 | 39.34 |
> >
> > [1] Model adaptation through hypothesis transfer with gradual knowledge distillation, In IROS21.
> >
> > [2] Guiding Pseudo-labels with Uncertainty Estimation for Source-free Unsupervised Domain Adaptation, In CVPR23.
> >
> > [3] Source-free domain adaptation via target prediction distribution searching, In IJCV24.

---

### Note · Authors · 2024-12-03

I have read and agree with the venue's withdrawal policy on behalf of myself and my co-authors.